# Learning Attribute-Aware Hash Codes for Fine-Grained Image Retrieval via Query Optimization

Peng Wang[1]  Yong Li[2]  Lin Zhao[1]  Xiu-Shen Wei[2†]

## Abstract

Fine-grained hashing has become a powerful solution for rapid and efficient image retrieval, particularly in scenarios requiring high discrimination between visually similar categories. To enable each hash bit to correspond to specific visual attributes, we propoe a novel method that harnesses learnable queries for attribute-aware hash codes learning. This method deploys a tailored set of queries to capture and represent nuanced attribute-level information within the hashing process, thereby enhancing both the interpretability and relevance of each hash bit. Building on this query-based optimization framework, we incorporate an auxiliary branch to help alleviate the challenges of complex landscape optimization often encountered with low-bit hash codes. This auxiliary branch models high-order attribute interactions, reinforcing the robustness and specificity of the generated hash codes. Experimental results on benchmark datasets demonstrate that our method generates attribute-aware hash codes and consistently outperforms state-of-the-art techniques in retrieval accuracy and robustness, especially for low-bit hash codes, underscoring its potential in fine-grained image hashing tasks. Our code is available at https://github.com/SEU-VIPGroup/QueryOpt

[1]School of Computer Science and Engineering, Nanjing University of Science and Technology, Nanjing, China. [2]School of Computer Science and Engineering, and Key Laboratory of New Generation Artificial Intelligence Technology and Its Interdisciplinary Applications, Southeast University, Nanjing, China. This work was uported by National Key R&D Program of China (2021YFA1001100), National Natural Science Foundation of China under Grant (62272231, 62172222), CIE-Tencent Robotics X Rhino-Bird Focused Research Program, the Fundamental Research Funds for the Central Universities (4009002401,4009002509), and the Big Data Computing Center of Southeast University. Correspondence to: Xiu-Shen Wei <weixs.gm@gmail.com>.

*Proceedings of the 42nd International Conference on Machine Learning*, Vancouver, Canada. PMLR 267, 2025. Copyright 2025 by the author(s).

## 1. Introduction

Large-scale fine-grained image retrieval aims to retrieve images from various subcategories within a meta-category, such as different species of animals, plant varieties, car models, or types of retail products (Wah et al., 2011; Horn et al., 2017; Wei et al., 2019). With the explosive growth of fine-grained data in real applications, fine-grained hashing has emerged as a promising solution for large-scale retrieval tasks (Wang et al., 2018; Cakir et al., 2018; Lin et al., 2015; Hu et al., 2024; Li et al., 2025), as hashing method significantly reduces storage costs and increases retrieval speed by using compact binary hash code representations. Unlike fine-grained classification, retrieval task face an open-world problem with unlimited categories. This task is complicated by subtle inter-class variations among similar subcategories and intra-class variations caused by different object postures (Wei et al., 2021b; Zhao et al., 2017; Wei et al., 2017). To address these challenges, it is essential to explore the nuances in object characteristics and leverage high-level fine-grained features.

Recently, a variety of learning-to-hash methods have been developed to enhance retrieval performance. In the literature, DSaH (Jin et al., 2020), ExchNet (Cui et al., 2020), SEMICON (Shen et al., 2022) and AGMH (Lu et al., 2023) concentrated on the design of feature extraction modules. As for the process of generating hash codes, the final hash codes are derived by projecting intricately coupled image features through a simple linear layer. Although the generated hash codes can perform retrieval tasks, a single bit will not imply significant semantics and it is difficult to preserve all the intricately coupled global features of an image. Additionally, $A^2$-NET (Wei et al., 2021a) and $A^2$-NET$^{++}$ (Wei et al., 2023a) employs a reconstruction task to constrain the process of generating hash codes, allowing each bit of hash codes to indicate attribute-level information. This nuanced attribute-level information enables us to effectively identify different categories. Consequently, hash codes can perform retrieval tasks effectively while ensuring that each bit is interpretable. Following the idea of utilizing attribute-level information as hash codes, we propose a novel method to generate attribute-aware hash codes for fine-grained image retrieval.

In this paper, we model the hash problem as a set prediction problem, where each element in the set represents a bit of the hash codes that is able to indicate a specific visual attribute. By structuring hash problem in this manner, an image can be directly decomposed into a set of attribute-specific features through learnable queries. Then each well-decomposed attribute-specific features is compressed into a final bit of hash codes for retrieval. Building upon this core idea, our proposed query optimization method integrates query learning to generate attribute-aware hash codes. Additionally, we incorporate an auxiliary branch for better optimization, further improving retrieval performance.

For query learning, a set of randomly initialized learnable queries is used for querying diverse attributes. Through a decoder based on cross-attention mechanism, these learnable queries interact with the extracted global features to directly decouple different attribute-specific features and then produce bits containing visual attributes level information. This direct querying method also demonstrates effectiveness and accuracy in other vision-related tasks (Carion et al., 2020; Kirillov et al., 2023; Jaegle et al., 2021). However, with the aforementioned process, we observe that the directly query learning strategy suffers from severely poor retrieval performance in low-bit hash codes scenarios. We analyze this phenomenon from the perspective of cosine similarity and discover the intrinsic reason lies in the inherent limitation of large class Numbers with low feature dimensions, resulting a challenges of complex landscape during the optimization process. This limitation makes it difficult for the model to learn distinguishable features. Based on the analysis, we employ an auxiliary branch only for training to help alleviate this challenges without introducing additional parameters. With this strategy, the model's performance is significantly enhanced, especially in low-bit scenarios. To evaluate our models, we conduct extensive experiments that provide both quantitative and qualitative results, and we also offer further experimental analysis of our method. The contributions of this work can be concluded as:

- We tailor the prevalent query learning mechanism to generate attribute-aware hash codes for fine-grained image retrieval tasks.

- We explain from the perspective of cosine similarity why retrieval performance is poor in low-bit hash codes scenarios. Based on our analysis, we seamlessly incorporate an auxiliary branch with query learning mechanism, which significantly improved the model's performance.

- Experiments conducted on five commonly used benchmark datasets for fine-grained image retrieval illustrate the effectiveness of our proposed method.

## 2. Related Work

**Fine-Grained Deep Hashing**    Hashing-based retrieval is a representative method for Approximate Nearest Neighbor Search (Wang et al., 2018; Liu et al., 2011; Li et al., 2013; Ye et al., 2022). Specifically, FPH (Yang et al., 2019) and DSaH (Jin et al., 2020) were among the earliest methods to incorporate hashing into fine-grained image retrieval. FPH introduced a novel two-pyramid hashing architecture to learn both semantic information and subtle appearance details. DSaH for the first time introduced the attention mechanism to the learning of fine-grained hashing codes. During the same period, ExchNet (Cui et al., 2020) investigated the large-scale fine-grained hashing task, proposing a unified end-to-end trainable network that captures both local and global features, representing parts and wholes of fine-grained objects. $A^2$-NET (Wei et al., 2021a) employed a similar localization module and attempted to learn high-level attribute vectors for hash code generation; its enhanced version, $A^2$-NET$^{++}$ (Wei et al., 2023a), further boosts the model's self-consistency. SEMICON (Shen et al., 2022) proposed a suppression-enhancing mask to explore the relationships between discovered regions. It generates the final hash codes in a stage-by-stage manner based on features from different levels, rather than aggregating features from different levels to generate unified hash codes. Most recently, AGMH (Lu et al., 2023) introduced an attention dispersion loss and a step-wise interactive external attention module to group and embed category-specific visual attributes in multiple descriptors for comprehensive feature representation. In this paper, we primarily follow the setup proposed by DPSH (Li et al., 2016b). Other notable works, such as FISH (Chen et al., 2022), CNET (Zeng & Zheng, 2023), DAHNET (Jiang et al., 2024) and CMBH (Chen et al., 2024) utilized classification tasks to enhance feature representation, which differ from the pairwise supervision setup and are not the primary focus of this paper.

**Set Prediction and Parallel Decoding**    The basic set prediction task is multilabel classification (Rezatofighi et al., 2017; Pineda et al., 2019; Li et al., 2016a). Furthermore, DETR (Carion et al., 2020) presented a new method that viewed object detection as a direct set prediction problem. It introduced an encoder-decoder transformer architecture that significantly simplified the object detection pipeline. Given a fixed, small set of learned object queries, DETR reasoned about the relationships among objects and the global image context to directly output the final set of predictions. Since the introduction of DETR, various modifications and improvements have been made (Zhu et al., 2021; Lin et al., 2022; Shehzadi et al., 2023; Chen et al., 2025), while the core concept of set prediction and the overall structure of the pipeline have remained intact. In our work, we treat the hashing problem as a set prediction task for distinguishable

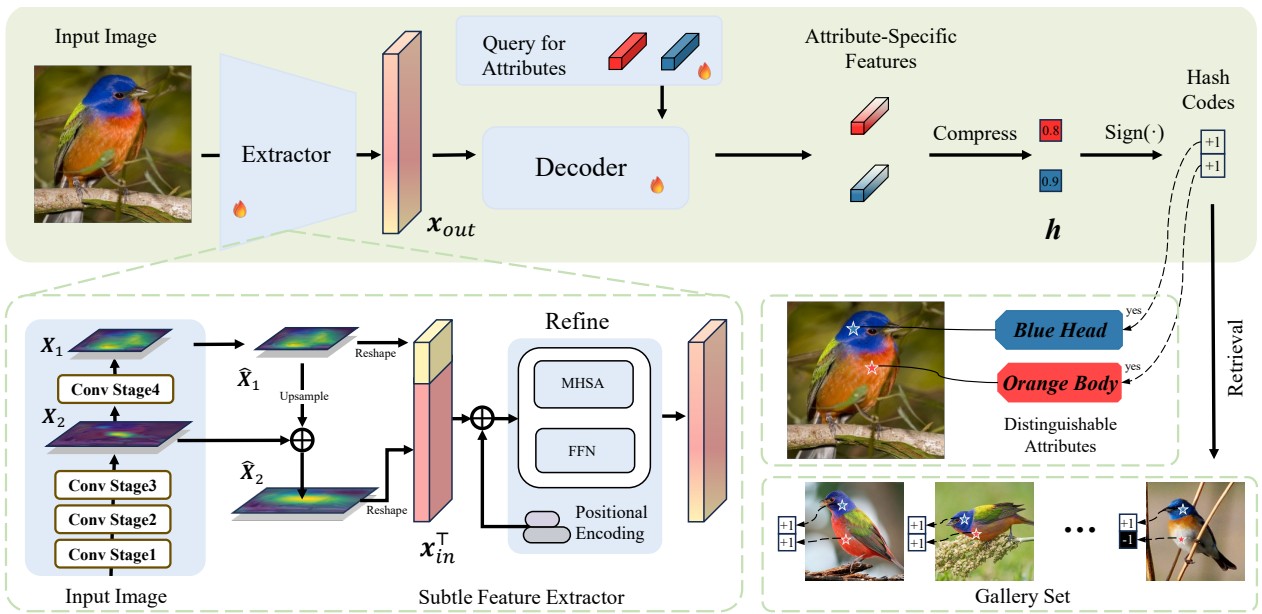

*Figure 1.* Overall framework of our method for generating attribute-aware hash codes for fine-grained image retrieval. The top part of the figure shows the workflow of the method: given a group of learnable parameters, an image is decoupled into a set of visual attributes, and each attribute is compressed into a single bit, serving as the hash code for retrieval. The bottom left part details the implementation of the Subtle Feature Extractor, where MHSA stands for multi-head self-attention and FFN stands for feed-forward network. The bottom right part illustrates our motivation for using distinctive attributes as hash codes for retrieval.

visual attributes which are more fine-grained than objects.

Parallel decoders are widely used in the field of deep learning, particularly in scenarios that require fast and efficient decoding, such as machine translation, text generation, and speech recognition (Vaswani et al., 2017; Chan et al., 2020). DETR (Carion et al., 2020) also combined with parallel decoding, achieving a suitable balance between the computational cost and the ability of performing global computations. The main advantage of parallel decoders was their ability to process multiple data points simultaneously, rather than sequentially, which greatly enhanced the speed and efficiency of decoding. For hash problems, parallel decoders handle the generation of variable-length hash codes, adding flexibility to the decoding process.

## 3. Methodology

Figure 1 illustrates the procedure for generating attribute-aware hash codes for an image, which is comprised of two key components: (1) a Subtle Feature Extractor (**S**, cf. Section 3.1) that obtains global image features for subsequent hash codes generation, and (2) a decoder with Query Learning mechanism (**Q**, cf. Section 3.2) to decouple the extracted global features into distinguishable attributes for the subsequent generation of attribute-aware hash codes. Additionally, we introduce an Auxiliary Branch (**A**, cf. Section 3.3) which reuses the aforementioned decoder for more effective attribute decoupling and performance enhancement. The

details are presented in the following sections.

### 3.1. Subtle Feature Extractor

In fine-grained tasks, images often contain extremely rich and subtle features (Gao et al., 2021; Haoran et al., 2023; Wei et al., 2023b). To obtain richer representations, and ensure that the decoding process is not hindered by this stage, we adopted two efficient strategies for subtle feature encoding, as shown in the bottom left part of Figure 1: (1) We constructed a top-down architecture to capture multi-scale features, thereby enabling the model to effectively capture the local characteristics of an image. (2) We incorporated self-attention mechanisms to eliminate irrelevant information, allowing the model to focus on identifying the most crucial regions.

Specifically, a raw pixel input image is represented as $\mathcal{I}_i \in \{\mathcal{I}_1, \ldots, \mathcal{I}_n\}$, where $n$ is the total number of training instances. We extract the deep feature of an input image $\mathcal{I}_i$ via a CNN backbone $\Phi_{\mathrm{CNN}}(\cdot)$ by:

$$\{\boldsymbol{X}_i^j | j \in \{1, \ldots, L\}\} = \Phi_{\mathrm{CNN}}(\mathcal{I}_i), \qquad (1)$$

where the feature map $\boldsymbol{X}_i^j \in \mathbb{R}^{c_j \times w_j \times h_j}$ is the output of the $j$th stage (from top to bottom) of the backbone. Then, we fuse the features by:

$$\hat{\boldsymbol{X}}_i^j = \begin{cases} \boldsymbol{X}_i^j & \text{if } j = 1, \\ \phi_{\mathrm{Conv}}(\boldsymbol{X}_i^j) + \phi_{\mathrm{up}}(\boldsymbol{X}_i^{j-1}) & \text{if } 1 < j \le L, \end{cases} \qquad (2)$$

where a $1 \times 1$ convolution layer $\phi_{\text{Conv}}$ adjusts the number of channels to $c_j$ and $\phi_{\text{up}}$ denotes upsampling operation.

To further refine the multi-scale features, a $1 \times 1$ convolution operation is first applied to reduce the channel dimension of the feature maps to $d$. The feature maps are then flattened and concatenated. Finally, $\{\hat{\boldsymbol{X}}_i^j | j \in \{1, \dots, L\}\}$ is transformed into $\boldsymbol{x}_{in} \in^{d \times E}$, where $E = \sum_{j=1}^{L} w_j \times h_j$. This sequence can then be processed by a standard transformer block. To improve computational efficiency, we only use a single layer of multi-head self-attention (MHSA) and a feed-forward network (FFN) to allow the model to focus on identifying significant regions.

Conclusively, the hybrid convolution and attention structure is utilized to enforce the model to focus on crucial fine-grained regions in the input image (Guo et al., 2022; Yang et al., 2021; Yu et al., 2024). The output $\boldsymbol{x}_{out} \in \mathbb{R}^{E \times d}$ will be further processed.

## 3.2. Query Optimization

**Query Learning**   Unlike other works that directly obtain hash codes through fully connected layers, our method leverages a decoder and a group of learnable queries to automatically decouple different attribute features from the global features $\boldsymbol{x}_{out}$. This decoupling process is illustrated at the middle top part of Figure 1. The automation in this process refers to the fact that it relies solely on the supervision signal indicating whether image pairs are similar or not, rather than explicit local attributes labeling information. Our query learning process decouples attribute-level features from intricately coupled global features. Then, each resulting attribute-specific features are compressed into one dimension and considered as a bit of hash codes. The decoder is implemented through cross-attention mechanism. To effectively decode diverse attribute-specific features, we randomly initialize $k$ different learnable attribute queries $\{\boldsymbol{q}_i \in \mathbb{R}^d | i \in \{1, \dots, k\}\}$ to interact with $\boldsymbol{x}_{out}$ through the decoder. This random initialization ensures the diversity of the queries, allowing queries to naturally focus on different features at the beginning of the learning process. This decoding process can be formulated as:

$$
\begin{aligned}
\boldsymbol{a}_i^m =& \text{softmax}\left( \frac{(\boldsymbol{q}_i^\top \boldsymbol{W}_q)^m (\hat{\boldsymbol{x}}_{out} \boldsymbol{W}_k)^{m^\top}}{\sqrt{d'}} \right) (\boldsymbol{x}_{out} \boldsymbol{W}_v)^m, \\
\boldsymbol{a}_i =& \text{Cat}(\boldsymbol{a}_i^1, \dots, \boldsymbol{a}^m) \boldsymbol{W}_o,
\end{aligned}
\tag{3}
$$

where $\boldsymbol{q}_i^\top \in \mathbb{R}^{1 \times d}$ represents one learnable query, $d'$ is the dimension of each head, $m = 1, \dots, M$ indices the output from the $m$-th attention head, and $\boldsymbol{W}_q, \boldsymbol{W}_k, \boldsymbol{W}_v \in \mathbb{R}^{d \times d}$ represent the learnable projection matrices. Additionally, $\hat{\boldsymbol{x}}_{out}$ denotes the position-augmented $\boldsymbol{x}_{out}$. We concatenate the outputs of each heads and using projection $\boldsymbol{W}_o$ to get the

distinguishable attribute-specific features $\boldsymbol{a}_i \in \mathbb{R}^{1 \times d}$ which can be compressed into $h_i = \frac{\boldsymbol{a}_i}{\|\boldsymbol{a}_i\|} \boldsymbol{W}$, where $\boldsymbol{W} \in \mathbb{R}^{d \times 1}$. Note that we use $\boldsymbol{Q} = [\boldsymbol{q}_1, \boldsymbol{q}_2, \dots, \boldsymbol{q}_k]^\top \in \mathbb{R}^{k \times d}$ to represent $k$ learnable queries and abstract the aforementioned process as a function $\boldsymbol{h} = H(\mathcal{I}_i, \boldsymbol{Q}, \boldsymbol{\Theta}) \in \mathbb{R}^k$, where the vector $\boldsymbol{h}$ is associated with parameters $\boldsymbol{Q}$ and $\boldsymbol{\Theta}$.

**Optimization Objective**   The optimization objective is defined based on the vector $\boldsymbol{h}$ that have been obtained. Assuming that we have $p$ test data points denoted as $\{\boldsymbol{p}_i\}_{i=1}^p$ and $d$ gallery set points denoted as $\{\boldsymbol{d}_j\}_{j=1}^d$. For each attribute vector $\boldsymbol{p}_i$ and $\boldsymbol{d}_j$, the corresponding hash codes can be generated by:

$$
\boldsymbol{v}_i = \text{sign}(\boldsymbol{p}_i), \quad \boldsymbol{z}_j = \text{sign}(\boldsymbol{d}_j).
\tag{4}
$$

In order to learn hash codes whose Hamming distances are minimized on similar image pairs and simultaneously maximized on dissimilar image pairs, we can minimize the $\ell_2$ loss between the pairwise supervised information and the inner product of query-database binary code pairs. Following (Jiang & Li, 2018), the formulation of hash code learning can be expressed through inner products as:

$$
\min_{\boldsymbol{\Theta}, \boldsymbol{Q}} \mathcal{L}(\mathcal{I}) = \sum_{i \in \Omega} \sum_{j \in \Gamma} \left[ \frac{\boldsymbol{v}_i^\top \boldsymbol{z}_j}{k} - S_{ij} \right]^2,
\tag{5}
$$

where $\boldsymbol{v}_i = \text{sign}(\boldsymbol{h})$ and $\Gamma$ presents the indices of all the gallery set points while $\Omega \subseteq \Gamma$ presents the indices of the train points for we can only gain access to a subset of gallery set points. $S_{ij} \in \{0, 1\}$ represents the pairwise supervised information, where $S_{ij} = 1$ if images $i$ and $j$ belong to the same category, and $0$ otherwise (Leng et al., 2014).

However, the gradient can not be back-propagated owing to the sign function. Hence, we substitute $\text{sign}(\cdot)$ for $\tanh(\cdot)$ to relax the whole optimization process. Typically, converting continuous codes $\boldsymbol{v}$ to binary codes $\boldsymbol{b}$ will lead to information loss, which is also known as quantization error. Therefore, we introduce a quantization loss to mitigate the impact of the relaxation. The loss of query optimization can be reformulated as:

$$
\mathcal{L}(\mathcal{I}) = \beta \sum_{i \in \Omega} \sum_{j \in \Gamma} \left[ \frac{\boldsymbol{v}_i^\top \boldsymbol{z}_j}{k} - S_{ij} \right]^2 + \gamma \sum_{i \in \Omega} [\boldsymbol{z}_i - \boldsymbol{v}_i]^2,
\tag{6}
$$

where $\boldsymbol{v}_i = \tanh(\boldsymbol{h})$.

## 3.3. Auxiliary Branch

**Observations of Poor Performance in Low-Bit Scenarios** We conducted experiments according to the above process. As shown in Figure 2, the comparison with the baseline demonstrates the effectiveness of our method. However, our

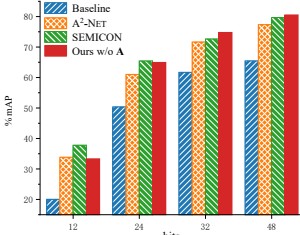

Figure 2. Comparison of our method (without an auxiliary branch $\boldsymbol{A}$, cf. Section 3.3) with two previous methods and baseline on *CUB200*.

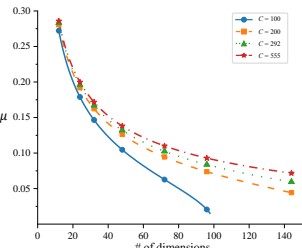

Figure 3. Lower bound of $\mu$ with different values of $C$.

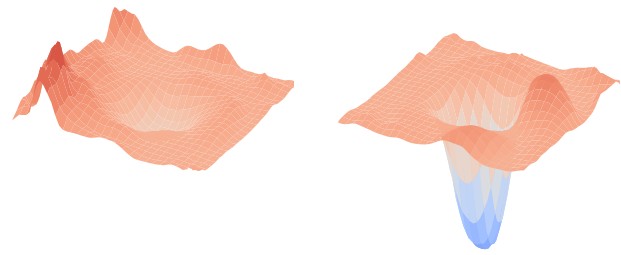

(a) # of dimensions = 12        (b) # of dimensions = 48

Figure 4. The loss landscape under $C = 200$.

method performs significantly worse in low-bit scenarios compared to the two previous state-of-the-art methods. Typically, it is assumed that low-bit hash codes perform poorly due to their limited bit length, which restricts the amount of information they can represent. Nevertheless, we indicate that this poor performance is primarily due to optimization difficulties associated with low-bit hash codes. In the following, we provide both a detailed analysis of this issue and the corresponding visualization results.

**Conjecture on the Limitation of Large Class Numbers with Low Feature Dimensions**   Geometrically, for $\boldsymbol{b} \in \{-1, +1\}^k$, the dot product can be represented by cosine similarity as:

$$\boldsymbol{b}_i^\top \boldsymbol{b}_j = \|\boldsymbol{b}_i\|\|\boldsymbol{b}_j\| \cos \alpha_{ij} = k \cos \alpha_{ij} \,, \quad (7)$$

where both $\|\boldsymbol{b}_i\|$ and $\|\boldsymbol{b}_j\|$ are constant (i.e., $\|\boldsymbol{b}\| = \sqrt{k}$). This means that we can analyze the loss function from the perspective of cosine similarity in continuous Euclidean space (Hoe et al., 2021). In designing our optimization objective, we actually facilitate distinguishable feature learning by minimizing the squared cosine similarity between the relaxed hash codes of pairs of samples from different image categories. Theoretically, in an $n$-dimensional space, there can be up to $n$ mutually orthogonal vectors. However, in practical optimization processes, the number of categories $C$ exceeds the spatial dimensions $n$, employing Stochastic Gradient Descent to minimize the sum of squared cosine similarities between all pairs becomes particularly challenging. Under such circumstances, we generally achieve a local

minimum that is greater than zero, rather than the ideal zero.

What kind of minimum can we optimize to achieve? We can randomly select one image from each category of training samples to compute its corresponding relaxed hash codes with $\tanh(\cdot)$. Specifically, we assemble the selected codes into a matrix $\boldsymbol{V} \in \mathbb{R}^{n \times C}$, where $\boldsymbol{V} = [\boldsymbol{v}_1, \boldsymbol{v}_2, ..., \boldsymbol{v}_C]$. We can define:

$$\mu = \max_{1 \le i, j \le N, i \ne j} \frac{|\boldsymbol{v}_i^\top \cdot \boldsymbol{v}_j|}{\|\boldsymbol{v}_i\|\|\boldsymbol{v}_j\|} = \max_{1 \le i, j \le N, i \ne j} \cos \alpha_{ij} \,. \quad (8)$$

When $C$ exceeds $n$, $\mu$ has a definite lower bound (Welch, 1974) can be formulated as:

$$\mu \ge \sqrt{\frac{C - n}{n(C - 1)}} \,, \quad (9)$$

where $\mu$ reflects the minimum state of the loss value during the optimization process. Since the loss function can be interpreted from the perspective of cosine similarity as discussed earlier, if $\mu$ is too large (i.e., $\cos \alpha_{ij}$ could potentially tend to approach 1), the model may incorrectly perceive samples from different classes as being similar. This makes it difficult for the model to learn distinguishable features. As shown in Figure 3, when $n$ increases from 12 to 48, the minimum value of $\mu$ decreases rapidly. Due to the rapid decrease of $\mu$ in higher-dimensional spaces, the cosine similarity between samples of different categories can be kept at a lower level, enabling the model to learn more distinguishable features. In addition, we present the visualization results of the loss landscape using the method from (Li et al., 2018) in Figure 4, which clearly show that in higher dimensions, achieving lower loss values is much easier.

Based on the above analysis, we conduct query transformation and add an auxiliary branch as a solution.

**Query Transformation and Auxiliary Branch for Better Optimization**   During the training process, our objective is to obtain well-optimized parameter $\{\boldsymbol{q}_i \in \mathbb{R}^d | i \in$

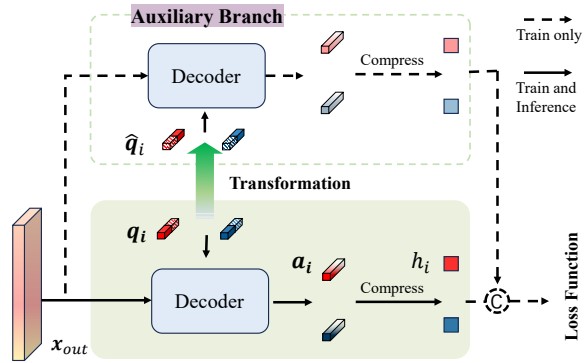

*Figure 5.* The $N - 1$ auxiliary branch with query transformation strategy (illustrated in the figure with $N = 2$ for simplicity, where $N$ is a hyperparameter) is used only during training.

$\{1, \ldots, k\}\}$ for effectively decoupling fine-grained attributes. With this objective and the motivation to conduct the learning process in a higher dimension to mitigate the analyzed limitations above, we conduct query transformation for each learnable query. We evenly slice the query vector and then perform a circular shift operation on the sliced sub-vectors. The shifted $\hat{q}_i$, which shares the same group of parameters with $q_i$, is processed with an auxiliary branch. We concatenate the output of both the original branch and the auxiliary branch for optimization. As illustrated in Figure 5, with one auxiliary branch, the numbers of dimensions is doubled. This query transformation and the incorporation of auxiliary branch offers two significant benefits: (1) the learnable queries interact more thoroughly with the enriched image features, and (2) the number of dimensions is increased without introducing any additional learnable parameters during training, which helps alleviate the challenges of complex landscape optimization encountered in low dimensions.

Mathematically speaking, given a learnable query $q_i \in \mathbb{R}^d$, we evenly split it into $N$ equal-length sub-vectors $[q_i^1; q_i^2; q_i^3; \ldots; q_i^N]$, then we perform a circular shift operation on the sliced sub-vectors to augment the new query:

$$\hat{q}_i^j = [q_i^{N-j+1}; \ldots; q_i^N; q_i^1; \ldots; q_i^{N-j}] \in \mathbb{R}^d, \quad (10)$$

where $j = 1, \ldots, N - 1$.

As illustrated in Figure 5, in the auxiliary branch, the vanilla decoder is reused to generate auxiliary outputs. During inference stage, we only retain the original $q_i$ during inference to generate $k$ bits hash codes.

### 3.4. Out-of-Sample Extension

After the training phase, we obtain the well-learned queries and other modules. Discarding the auxiliary branch, our method can be employed to generate hash codes for an input image $\mathcal{I}_i$ as follows:

$$u_i = \text{sign}(H(\mathcal{I}_i, Q, \Theta)). \quad (11)$$

## 4. Experiment

**Datasets** We conducted a series of ablation studies and comparisons with the latest methods on five fine-grained datasets, i.e., *CUB200* (Wah et al., 2011), *Aircraft* (Maji et al., 2013), *Food101* (Bossard et al., 2014), *VegFru* (Hou et al., 2017) and *NABirds* (Van Horn et al., 2015). During the evaluation, we used the official training and testing sets as the retrieval and query sets. Specifically, *CUB200* contains 11,788 bird images from 200 bird species and is officially split into 5,994 images for training and 5,794 images for testing. *Aircraft* includes 10,000 images of 100 aircraft variants, with 6,667 images designated for training and 3,333 for testing. For large-scale datasets, *Food101* features 101 kinds of foods with 101,000 images, each class has 250 test images that are manually checked for correctness, while the 750 training images may still contain a certain amount of noise. *NABirds* comprises 48,562 images of north american birds across 555 sub-categories, with 23,929 images for training and 24,633 for testing. *VegFru* is another large-scale fine-grained dataset covering 200 kinds of vegetables and 92 kinds of fruits, with 29,200 images for training, 14,600 for validation, and 116,931 for testing.

**Implementation Details** For fair comparisons of fine-grained hashing, we follow the efficient training setting outlined in ExchNet (Cui et al., 2020). Concretely, for *CUB200*, *Aircraft* and *Food101*, we sample 2,000 images per epoch, while 4,000 samples are randomly selected for *NABirds* and *VegFru*. Following the method of ExchNet, ResNet-50 (He et al., 2016) is employed in experiments. The number of total training epoch is the same as SEMICON (Shen et al., 2022) and AGMH (Lu et al., 2023).

For all datasets, we preprocess all images to $224 \times 224$. Initial learning rate is 0.0003. SGD with a mini-batch size of 16 is used for training. We set the weight decay to 0.0001 and momentum to 0.90. We set $L = 2$ to leverage multi-scale features. $d$ is set to 384, and $N$ is set to $\frac{96}{k}$. All experiments are conducted with one GeForce RTX 3090 GPU.

**Comparison Methods** To prove the superiority of our method, we compare it with existing hashing-based methods. Among them, DPSH (Li et al., 2016b), HashNet (Cao et al., 2017) and ADSH (Jiang & Li, 2018) are coarse-grained methods. ExchNet (Cui et al., 2020), A$^2$-NET (Wei et al., 2021a), SEMICON (Shen et al., 2022), A$^2$-NET$^{++}$ (Wei et al., 2023a) and AGMH (Lu et al., 2023) are advanced hashing methods aiming at large-scale fine-grained retrieval.

Table 1. Comparisons of retrieval accuracy (% mAP) on five fine-grained benchmark datasets.

| Datasets | # bits | DPSH | HashNet | ADSH | ExchNet | $A^2$-NET | SEMICON | $A^2$-NET$^{++}$ | AGMH | Ours |
|---|---|---|---|---|---|---|---|---|---|---|
| CUB200 | 12 | 8.68 | 12.03 | 20.03 | 25.14 | 33.83 | 37.76 | 37.83 | 56.42 | **72.19** |
|  | 24 | 12.51 | 17.77 | 50.33 | 58.98 | 61.01 | 65.41 | 71.73 | 77.44 | **81.38** |
|  | 32 | 12.74 | 19.93 | 61.68 | 67.74 | 71.61 | 72.61 | 78.39 | 81.95 | **82.22** |
|  | 48 | 15.58 | 22.13 | 65.43 | 71.05 | 77.33 | 79.67 | 82.71 | 83.69 | **83.96** |
| Aircraft | 12 | 8.74 | 14.91 | 15.54 | 33.27 | 42.72 | 49.87 | 57.53 | 71.64 | **78.47** |
|  | 24 | 10.87 | 17.75 | 23.09 | 45.83 | 63.66 | 75.08 | 73.45 | 83.45 | **83.88** |
|  | 32 | 13.54 | 19.42 | 30.37 | 51.83 | 72.51 | 80.45 | 81.59 | 83.60 | **84.06** |
|  | 48 | 13.94 | 20.32 | 50.65 | 59.05 | 81.37 | 84.23 | **86.65** | 84.91 | 85.60 |
| Food101 | 12 | 11.82 | 24.42 | 35.64 | 45.63 | 46.44 | 50.00 | 54.51 | 62.59 | **70.69** |
|  | 24 | 13.05 | 34.48 | 40.93 | 55.48 | 66.87 | 76.57 | 81.46 | 80.94 | **81.76** |
|  | 32 | 16.41 | 35.90 | 42.89 | 56.39 | 74.27 | 80.19 | 82.92 | 82.31 | **82.74** |
|  | 48 | 20.06 | 39.65 | 48.81 | 64.19 | 82.13 | 82.44 | 83.66 | 83.21 | **83.81** |
| NABirds | 12 | 2.17 | 2.34 | 2.53 | 5.22 | 8.20 | 8.12 | 8.80 | – | **28.13** |
|  | 24 | 4.08 | 3.29 | 8.23 | 15.69 | 19.15 | 19.44 | 22.65 | – | **37.75** |
|  | 32 | 3.61 | 4.52 | 14.71 | 21.94 | 24.41 | 28.26 | 29.79 | – | **41.33** |
|  | 48 | 3.20 | 4.97 | 25.34 | 34.81 | 35.64 | 41.15 | 42.94 | – | **43.71** |
| VegFru | 12 | 6.33 | 3.70 | 8.24 | 23.55 | 25.52 | 30.32 | 30.54 | 43.99 | **69.76** |
|  | 24 | 9.05 | 6.24 | 24.90 | 35.93 | 44.73 | 58.45 | 60.56 | 68.05 | **83.30** |
|  | 32 | 10.28 | 7.83 | 36.53 | 48.27 | 52.75 | 69.92 | 73.38 | 76.73 | **83.81** |
|  | 48 | 9.11 | 10.29 | 55.15 | 69.30 | 69.77 | 79.77 | 82.80 | 84.49 | **85.38** |

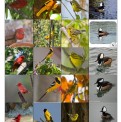 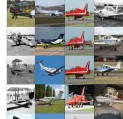 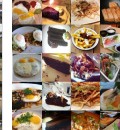 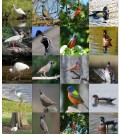 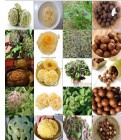

(a) CUB200  (b) Aircraft  (c) Food101  (d) NABirds  (e) VegFru

Figure 6. Quality demonstrations of the learned attribute hash codes. Each column in each sub-figure can strongly correspond to a certain kind of properties of the fine-grained objects, e.g., "red birds", "double-wing aircrafts", "fried eggs", "long beak" "layered petal-like structure", etc. (Best viewed in color and zoomed in.)

We provide a comparison with classification-based methods, such as FISH (Chen et al., 2022), CNET (Zeng & Zheng, 2023), DAHNET (Jiang et al., 2024) and CMBH (Chen et al., 2024), in appendix.

### 4.1. Quantitative Results

Table 1 presents the mean average precision (mAP) results for fine-grained retrieval across five benchmark datasets. For each dataset, we evaluate performance using four different hash codes lengths: 12, 24, 32 and 48. As shown in Table 1, our method consistently outperforms the previous methods across all datasets. It's important to highlight the factor that contribute to our method's superior performance. By alleviating optimization challenges in

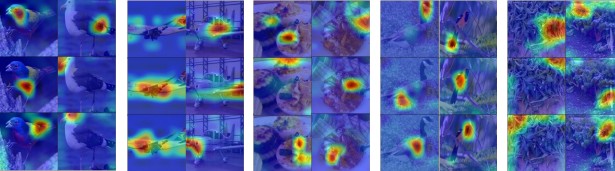

(a) CUB200  (b) Aircraft  (c) Food101  (d) NABirds  (e) VegFru

Figure 7. Visualization of different queries. Each row corresponds to a distinct learnable attribute query ($q_i \in \mathbb{R}^d$). (Best viewed in color and zoomed in.)

low-bit hash codes scenarios, we significantly improve retrieval performance. Thereby narrowing the performance gap across various lengths of hash codes. Notably, in the 12-bits setting, our method shows substantial enhancements on the CUB200 and VegFru datasets, achieving 15.77% and 25.77% improvements over AGMH (Lu et al., 2023), respectively. This enhancement is especially valuable for practical applications, where fast retrieval and reduced storage requirements are essential. In contrast to other previous methods, $A^2$-NET (Wei et al., 2021a) and $A^2$-NET$^{++}$ (Wei et al., 2023a) enhance learned bits by incorporating strong visual features through reconstruction operations. Our query optimization, as demonstrated in Section 4.2 and Figure 6, not only ensures the learned hash codes maintain the advantage of attribute awareness but also achieves superior retrieval performance.

*Table 2.* Retrieval accuracy (% mAP) with incremental components.

| Configurations | | | CUB200 | | | |
|---|---|---|---|---|---|---|
| S | Q | A | 12bits | 24bits | 32bits | 48bits |
| ✗ | ✗ | ✗ | 20.03 | 50.33 | 61.68 | 65.43 |
| ✔ | ✗ | ✗ | 27.43 | 59.19 | 67.69 | 77.07 |
| ✔ | ✔ | ✗ | 33.33 | 64.94 | 74.79 | 80.52 |
| ✔ | ✔ | ✔ | **72.19** | **81.38** | **82.22** | **83.96** |

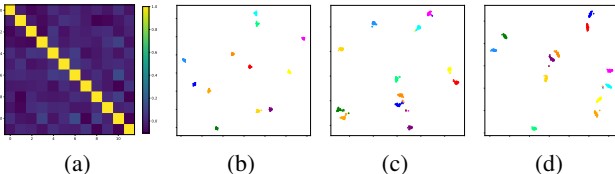

| (a) | (b) | (c) | (d) |

*Figure 8.* Visualization results of learned queries with the 12-bits setting. (a) is the similarity matrix of 12 queries. (b), (c) and (d) are the $t$-SNE (van der Maaten & Hinton, 2008) results of the input images from three different categories, respectively. Different colors represent different attribute-specific features.

## 4.2. Qualitative Results

We hereby discuss the quality of the attribute-aware hash codes generated by our method. Similar to the method of A$^2$-NET (Wei et al., 2021a) and A$^2$-NET$^{++}$ (Wei et al., 2023a), we visualize the retrieved images using a single bit randomly selected from the hash codes to highlight the strong link between visual attributes. All five datasets from our experiments serve as examples to demonstrate the quality. As depicted in Figure 6, images in each column exhibit similar fine-grained object properties, i.e., visual attributes. The learned hash codes clearly exhibits a strong attribute-awareness, which indicates that this direct attributes query method is interpretable. As can be seen from Figure 7, the activated regions of different queries (highlighted in warm colors) are also semantically meaningful. These visualizations qualitatively illustrate that our proposed query learning achieves strong interpretability.

## 4.3. Further Analysis

**Ablation Studies of Different Modules** We conduct ablation studies on *CUB200* to verify the effectiveness of each module, including a Subtle Feature Extractor (**S**, cf. Section 3.1), the Query Learning mechanism (**Q**, cf. Section 3.2), and the Auxiliary Branch (**A**, cf. Section 3.3). These modules are incrementally applied to a base network (i.e., ResNet-50). As illustrated in Table 2, by adding these modules one by one, the retrieval results are steadily improved, which justifies the effectiveness of our method. Based on the enriched features, the query learning mechanism can generating attribute-aware hash codes, thereby

*Table 3.* Comparison results of hyper parameter $N$. Results are based on *CUB200* and *Aircraft* datasets under the 12-bits setting.

| $N$ | 1 | 2 | 4 | 6 | 8 | 12 |
|---|---|---|---|---|---|---|
| *CUB200* | 33.33 | 52.44 | 69.46 | 71.57 | 72.19 | 72.36 |
| *Aircraft* | 39.43 | 65.70 | 74.51 | 75.20 | 78.47 | 78.38 |

*Table 4.* Parameters and computational costs (Flops) for training and inference. Methods with source codes publicly released are included. Results are based on 12-bits setting.

| Method | Params | Flops | |
|---|---|---|---|
| | | training | inference |
| ExchNet | 40.43M | 7.67G | 7.67G |
| SEMICON | 42.42M | 7.09G | 7.09G |
| AGMH | 35.56M | 8.61G | 8.61G |
| Ours | **31.71M** | **5.27G** | **5.23G** |

improving performance. Additionally, the incorporation of auxiliary branch mitigate the inherent limitations caused by the low dimensions, significantly improving performance.

**Attribute Diversity of Query Learning** In our method, the learnable queries are randomly initialized, which inherently introduces diversity without requiring additional constraints. At the beginning of the training process, this diversity naturally directs focus toward different attributes of the input images, and we illustrate the well-learned queries in Figure 8a with the 12-bit setting. The results indicate that the 12 well-learned queries are distinct, showing that this diversity is preserved throughout training.

Apart from the learnable queries themselves, a key question is whether our query mechanism can efficiently decouple visual attributes that exhibit diversity. We visualized these decoupled attribute-specific features in Figures 8b, 8c and 8d. The images from three different categories are represented by 12 different attributes, respectively. This demonstrates that our method efficiently decouples diverse attributes, ensuring that the same image is represented by 12 bits containing diverse semantic information.

**Number of Auxiliary Branches** We increased the number of dimensions during training to $N \times k$ by applying query transformation and adding $N - 1$ auxiliary branches. The retrieval results with different values of $N$ under the 12-bits setting are presented in Table 3. As shown, when $N$ increases from 1 to 6, retrieval performance improves rapidly, closely aligning with the sharp decline of $\mu$ shown in Figure 3. Once $N$ exceeds 6, the changes in $\mu$ become more gradual, and also the performance stabilizes. Finally, we set $N = \frac{96}{k}$, where 96 is the least common multiple of 12, 24, 32 and 48.

**Model Complexity and Computational Costs** A potential concern is whether our method sacrifices efficiency, especially when optimizing in high-dimensional spaces. However, as shown in Table 4, our method is more lightweight and computational efficient compared to some previous methods. Our strategy, which conducts query transformation and incorporates an auxiliary branch for better optimization, only introduces a negligible computational overhead.

## 5. Conclusion

In this paper, we proposed a novel method that can generate attribute-aware hash codes for large-scale fine-grained image retrieval task. Specifically, to obtain subtle features, we first extracted multi-scale information from a CNN backbone and then refined these features. Then, by utilizing a decoder based on cross-attention mechanisms and a group of learnable parameters which are treated as attribute queries, we directly decouple attribute-specific features and compress them into hash codes for retrieval. The generated attribute-aware hash codes were able to provide interpretability. Additionally, we explored the limitations between large class numbers and low optimization dimensions during the optimization process. We proposed a query transformation strategy and trained the model with an auxiliary branch, which significantly enhanced the model's performance without introducing any additional parameters. Furthermore, the decoder has the potential to be extended to interact with text embeddings, which encourages us to explore cross-modal hashing in future research.

## Impact Statement

This paper presents work whose goal is to advance the field of Machine Learning. There are many potential societal consequences of our work, none which we feel must be specifically highlighted here.

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

# A. Appendix Overview

In this appendix, we present additional information about our proposed method, including: (1) an explanation of enhanced interaction in the auxiliary branch; (2) further experimental results that compare our method with classification-based methods, examples of retrieved results, and additional visualization results; and (3) a proof for the lower bound of $\mu$.

# B. Explanation of Enhanced Interaction in Auxiliary Branch

One of the advantages of query transformation and the incorporation of an auxiliary branch is that the learnable queries interact more thoroughly with the enriched image features. As illustrated in Figure 9, both $q$ and $\hat{q}$ share the same parameters, however, their interactions with the features differ. By including the auxiliary branch, we effectively expand the receptive field of the learnable queries, enhancing the flexibility and comprehensiveness of their interactions, which enables the queries to capture complex patterns within the input images more effectively.

# C. Additional Experimental Result

**Comparison with Classification-Based Methods**  Apart from the methods mentioned in the main body, FISH (Chen et al., 2022), CNET (Zeng & Zheng, 2023), DAHNET (Jiang et al., 2024) and CMBH (Chen et al., 2024) are other fine-grained hashing methods that have achieved good retrieval accuracy. For fair comparisons, we control empirical settings to be the same as these methods and incorporate additional classification tasks into our method during the training phase. The comparison results are presented in Table 5, demonstrating that our method is comparable to these methods.

**Examples of Retrieved Results**  We present retrieval results on *CUB200-2011* (Wah et al., 2011) and *Food101* (Bossard et al., 2014). As shown in Figure 10 and Figure 11, our proposed method performs well in retrieving among multiple subordinate categories. However, there are several failure cases where minimal differences, such as those caused by different views or the presence of distracting objects, require careful observation between the query image and the returned images.

**Visualization of Loss Landscape**  We provide an extended analysis with additional visualization cases across multiple $C$ values ($C = 200, 555$) in Figure 12. The comprehensive results consistently reveal that:

- The loss landscape exhibits a strong correlation with,

*Table 5.* Comparison of Retrieval Accuracy (% mAP) with Classification-Based Methods.

| Dataset | CUB200-2011 | | | |
|---------|---------|---------|---------|---------|
| Methods | 12 bits | 24 bits | 32 bits | 48 bits |
| FISH | 76.77 | 79.93 | 80.09 | 80.88 |
| CNET | 77.10 | 82.11 | 83.09 | 83.92 |
| DAHNET | 61.69 | 79.00 | 81.69 | 83.98 |
| CMBH* | 80.30 | **82.16** | 82.68 | 82.80 |
| Ours[†] | **80.79** | 82.02 | **83.34** | **84.27** |

* denotes CMBH without multi-region feature embedding to ensure similar training costs, and [†] denotes our method is trained with classification.

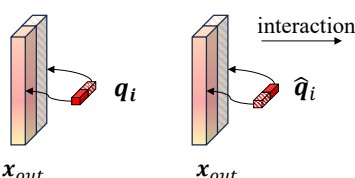

*Figure 9.* The interaction of learnable query with enriched image features.

aligning well with the trends shown in Figure 3.

- Larger class numbers coupled with lower feature dimensions indeed lead to inaccessible lower local minima and a less smooth loss landscape.

This phenomenon aligns with our conjecture on the limitation of large class numbers with low feature dimensions.

**Visualization of Extracted Features**  In Section 3.3, we conjecture that a large number of classes with low feature dimensions can make it difficult for the model to learn distinguishable features. To mitigate this limitation, we propose a strategy of adding an auxiliary branch. To validate our conjecture, we employ $t$-SNE to visualize the features $x_{out}$. The visualization results in figure 13 demonstrate that incorporating the auxiliary branch during training produces feature distributions exhibiting two key characteristics: (1) larger inter-class distances, manifested through more distinct cluster separations between categories; and (2) smaller intra-class variations, evidenced by tighter aggregation of samples within each category. In contrast, the model without the auxiliary branch produces features with substantially overlapping distributions across different categories. This quantitative evidence strongly supports our claim that the auxiliary branch enhances feature discriminability.

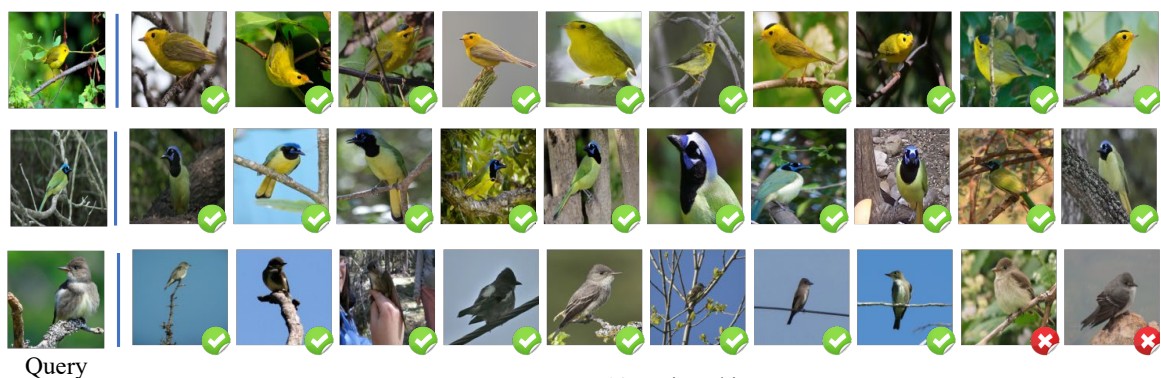

Query
image

Top-10 retrieved images

*Figure 10.* Examples of top-10 retrieved images on *CUB200-2011* of 48-bit hash codes by our method.

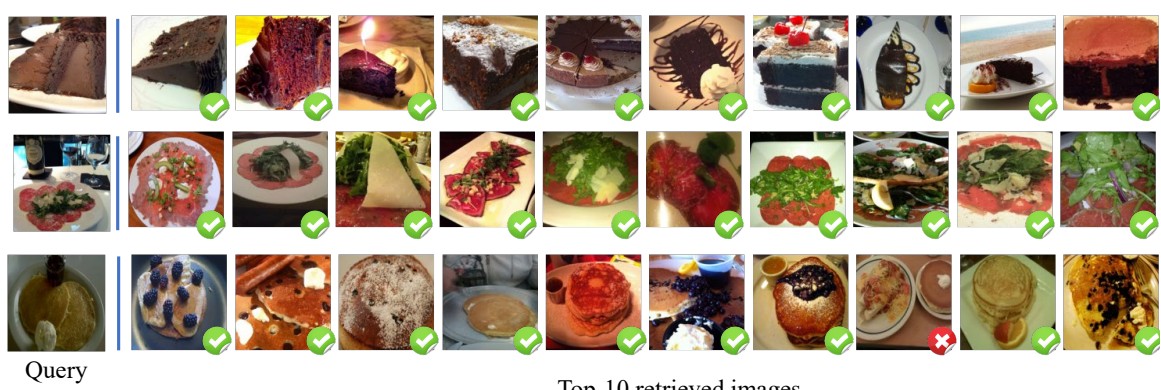

Query
image

Top-10 retrieved images

*Figure 11.* Examples of top-10 retrieved images on *Food101* of 48-bit hash codes by our method.

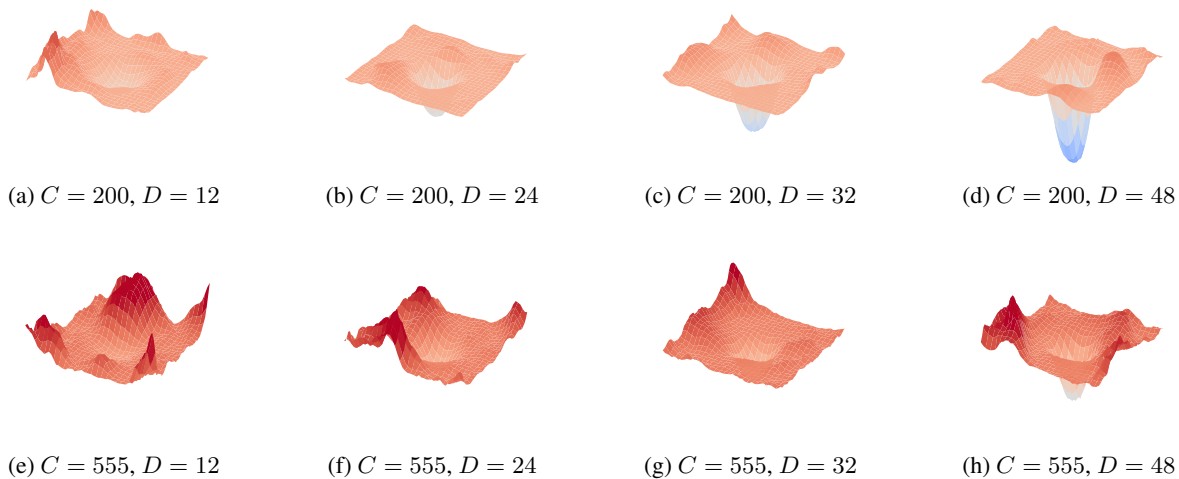

(a) $C = 200, D = 12$      (b) $C = 200, D = 24$      (c) $C = 200, D = 32$      (d) $C = 200, D = 48$

(e) $C = 555, D = 12$      (f) $C = 555, D = 24$      (g) $C = 555, D = 32$      (h) $C = 555, D = 48$

*Figure 12.* Loss landscape across different parameter configurations: (a-d) show results with $C = 200$, and (e-h) demonstrate cases with $C = 555$. Each column corresponds to specific dimension settings, where $C$ represents the number of classes and $D$ denotes the number of dimensions.

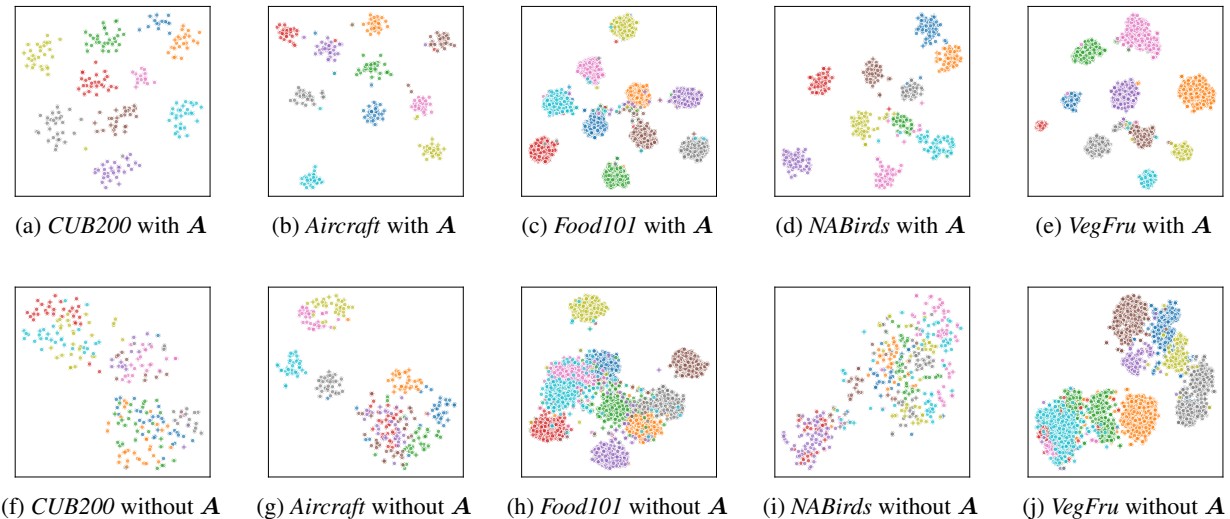

(a) *CUB200* with $\boldsymbol{A}$    (b) *Aircraft* with $\boldsymbol{A}$    (c) *Food101* with $\boldsymbol{A}$    (d) *NABirds* with $\boldsymbol{A}$    (e) *VegFru* with $\boldsymbol{A}$

(f) *CUB200* without $\boldsymbol{A}$    (g) *Aircraft* without $\boldsymbol{A}$    (h) *Food101* without $\boldsymbol{A}$    (i) *NABirds* without $\boldsymbol{A}$    (j) *VegFru* without $\boldsymbol{A}$

*Figure 13.* $t$-SNE Visualization of $\boldsymbol{X}_{out}$ for 10 randomly selected classes

## D. A Proof for the Lower Bound

In our conjecture on the limitation of large class numbers with low feature dimensions, we introduce a lower bound for analyzing the observations of poor performance in low-bit scenarios. This lower bound is also called the Welch bound. Referring to (Welch, 1974), we provide a proof.

We use the relaxed hash codes $\boldsymbol{v}_i = \tanh(H(\mathcal{I}_i, \boldsymbol{Q}, \boldsymbol{\Theta}))$. to construct a matrix $\boldsymbol{V} = [\boldsymbol{v}_1, \boldsymbol{v}_2, ..., \boldsymbol{v}_C] \in \mathbb{R}^{n \times C}$ with $C > n$. we first normalized the columns and let $\boldsymbol{x}_i = \frac{\boldsymbol{v}_i}{\|\boldsymbol{v}_i\|}$ get the correspond gram matirx:

$$\boldsymbol{G} = \begin{bmatrix} \langle \boldsymbol{x}_1, \boldsymbol{x}_1 \rangle & \cdots & \langle \boldsymbol{x}_1, \boldsymbol{x}_C \rangle \\ \vdots & \ddots & \vdots \\ \langle \boldsymbol{x}_C, \boldsymbol{x}_1 \rangle & \cdots & \langle \boldsymbol{x}_C, \boldsymbol{x}_C \rangle \end{bmatrix}, \quad (12)$$

where $\langle \cdot, \cdot \rangle$ is the usual inner product on $\mathbb{R}^n$.

The trace of $\boldsymbol{G}$ is equal to the sum of its eigenvalues. Because the rank of $\boldsymbol{G}$ is at most $n$, and is a positive semidefinite matrix, $\boldsymbol{G}$ has at most $n$ positive eigenvalues with its remaining eigenvalues all equal to zero. Writing the non-zero eigenvalues of $\boldsymbol{G}$ as $\lambda_1, \ldots, \lambda_r$, with $r \leq n$ and applying the Cauchy-Schwarz inequality we can get:

$$(\operatorname{Tr} \boldsymbol{G})^2 = \left( \sum_{i=1}^{r} \lambda_i \right)^2 \leq r \sum_{i=1}^{r} \lambda_i^2 \leq n \sum_{i=1}^{n} \lambda_i^2. \quad (13)$$

The square of the Frobenius norm of $\boldsymbol{G}$ satisfies:

$$\|\boldsymbol{G}\|_F^2 = \sum_{i=1}^{C} \sum_{j=1}^{C} (\langle \boldsymbol{x}_i, \boldsymbol{x}_j \rangle)^2 = \sum_{i=1}^{C} \lambda_i^2. \quad (14)$$

Taking Equation (13) and Equation (14) together with the preceding inequality gives:

$$\sum_{i=1}^{C} \sum_{j=1}^{C} (\langle \boldsymbol{x}_i, \boldsymbol{x}_j \rangle)^2 \geq \frac{(\operatorname{Tr} \boldsymbol{G})^2}{n}. \quad (15)$$

As $\boldsymbol{x}_i$ are unit vector, so the diagonal of $\boldsymbol{G}$ are all 1, hence $\operatorname{Tr} \boldsymbol{G} = C$ and

$$\sum_{i=1}^{C} \sum_{j=1}^{C} (\langle \boldsymbol{x}_i, \boldsymbol{x}_j \rangle)^2 = C + \sum_{i \neq j} (\langle \boldsymbol{x}_i, \boldsymbol{x}_j \rangle)^2. \quad (16)$$

Therefore,

$$\sum_{i \neq j} (\langle \boldsymbol{x}_i, \boldsymbol{x}_j \rangle)^2 \geq \frac{C(C - n)}{n}. \quad (17)$$

The mean of a set of non-negative numbers is smaller than their maximum:

$$\max(\langle \boldsymbol{x}_i, \boldsymbol{x}_j \rangle)^2 \geq \frac{1}{C(C - 1)} \sum_{i \neq j} (\langle \boldsymbol{x}_i, \boldsymbol{x}_j \rangle)^2. \quad (18)$$

So,

$$\mu \geq \sqrt{\frac{C - n}{n(C - 1)}}. \quad (19)$$