# OpenReview forum: "Learning Attribute-Aware Hash Codes for Fine-Grained Image Retrieval via Query Optimization"
_ICML.cc/2025/Conference — ICML 2025 poster_

### Official Review · Reviewer_Zq7Z · 2025-03-09

**Overall Recommendation:** 4

**Summary:**

This paper presents a novel learn-to-hash method for large-scale fine-grained image retrieval. It introduces a query learning mechanism that can capture nuanced attribute-level information, making each bit of the hash code interpretable. The paper also introduces auxiliary branches during the training process to improve model performance. Experiments on five datasets demonstrate the effectiveness of the method.

**Claims And Evidence:**

This paper analyzes the pairwise loss used in its method from the perspective of cosine similarity, arguing that when the number of categories exceeds the feature dimensions, the model may not learn distinguishable features. In addition to the theoretical analysis, the authors also provide visual results of the loss landscape, clearly illustrating that it is more difficult to achieve lower loss values when the number of dimensions is small.

**Essential References Not Discussed:**

No Essential References Not Discussed

**Experimental Designs Or Analyses:**

The experiments conducted on five fine-grained datasets, covering various scenarios such as birds, airplanes, and food, demonstrate that our method significantly outperforms previous approaches in low-bit scenarios. This underscores its broad applicability. Additionally, ablation study further validate the effectiveness of each module.

**Methods And Evaluation Criteria:**

I think the method proposed in the paper is meaningful for the application of fine-grained image retrieval.

**Other Comments Or Suggestions:**

There is a spelling error in line 302.

**Other Strengths And Weaknesses:**

Strengths:
1，This paper is well organized, and the figures and tables are presented clearly.
2，This paper addresses the important and practical task of large-scale fine-grained retrieval, presenting an effective method that enhances retrieval accuracy while also offering intriguing interpretability. The entire method is well-motivated.
3，This paper analyzes why retrieval performance is poor in low-bit hash code scenarios from the perspective of cosine similarity and proposes an efficient approach that introduces an additional auxiliary branch during the training process as a solution.
4,The authors provide many visualizations to help readers understand the method.

Weaknesses:
1. Figure 3 illustrates the curves at different values of c, and the authors further support their analysis with the visual results in Figure 4. However, Figure 4 only provides visualization for the specific case of C=200. Does the trend of the loss landscape at different values of c align with that in Figure 3? More visual results should be included.
2.Section 3.3 mentions that without the auxiliary branch, the model may not learn discriminative features. Are there any experimental results to support this observation? The authors could consider plotting the t-SNE result of X_out to observe the feature distribution across different categories.

**Questions For Authors:**

Please refer to Weaknesses.

**Relation To Broader Scientific Literature:**

This paper innovatively models the hash problem as a set prediction task inspired by DETR. By leveraging learnable queries, the method not only enhances interpretability but also demonstrates improved retrieval capabilities. This contribution explores the intersection of hashing techniques and cross attention mechanisms, offering a fresh perspective that could inspire further research in both hashing and attribute-based image retrieval.

**Theoretical Claims:**

This paper leverages auxiliary branch to increase the number of dimension, alleviating the low-bit optimization challenge. Experiments regarding the hyperparameter N show that the experimental results are consistent with the theoretical analysis.

---

> ### Author Rebuttal · Authors · 2025-03-29
>
> **Comment1:** *Figure 3 illustrates the curves at different values of c, and the authors further support their analysis with the visual results in Figure 4. However, Figure 4 only provides visualization for the specific case of C=200. Does the trend of the loss landscape at different values of c align with that in Figure 3? More visual results should be included.*
>
> **Reply1:**
> We sincerely appreciate your constructive feedback. In response to your insightful suggestion, we have supplemented the analysis with additional visualization cases across multiple c-values (C = 200, 292, 555). These extended results consistently demonstrate that:
>
> 1. The loss landscape exhibits a strong correlation with $\mu$, aligning well with the trends shown in Figure 3.
> 2. Larger class numbers coupled with lower feature dimensions indeed lead to:
>    - Inaccessible lower local minima
>    - Less smooth loss landscape
>
>
> This phenomenon aligns with our ​conjecture on the limitation of large class numbers with low feature dimensions. The visualization results can be found at: https://anonymous.4open.science/r/rebuttal_for_ICML2025-89B6/figure1.png
>
> ---
>
> **Comment2:** *Section 3.3 mentions that without the auxiliary branch, the model may not learn discriminative features. Are there any experimental results to support this observation? The authors could consider plotting the t-SNE result of X_out to observe the feature distribution across different categories.*
>
> **Reply2:**
> We sincerely appreciate your insightful suggestion. To validate our observation, we have conducted t-SNE visualization on the feature embeddings $X_{out}$. The results demonstrate that when training ​with the auxiliary branch, the feature distributions exhibit:
> 1. ​Larger inter-class distances: Distinct categories form more separated clusters.
> 2. ​Smaller intra-class variations: Samples within the same category show tighter aggregation.
>
> In contrast, the model ​without the auxiliary branch produces features with substantially overlapping distributions across different categories. This quantitative evidence strongly supports our claim that the auxiliary branch enhances feature discriminability. visualization results can be found at: https://anonymous.4open.science/r/rebuttal_for_ICML2025-89B6/figure2.png
>
>
> ---
>
> **Comment3:** *There is a spelling error in line 302.*
>
> **Reply3:**
> We sincerely appreciate your thorough review. We have carefully addressed the identified issues and conducted further proofreading of the manuscript to ensure its quality.

---

> > ### Comment · Reviewer_Zq7Z · 2025-04-02
> >
> > I am satisfied with the author's rebuttal as it has effectively addressed my concerns. Therefore, I would like to increase my score to 4.

---

### Official Review · Reviewer_gSZJ · 2025-03-09

**Overall Recommendation:** 3

**Summary:**

In this paper, the authors propose a query optimization-based fine-grained image hashing method, which enables the generated hash bits to exhibit attribute-aware characteristics. From the perspective of cosine similarity, the challenges in generating effective low-bit hash codes are analyzed. Based on this analysis, the performance of the model is particularly enhanced for the low-bit case through the incorporation of auxiliary branches. Expensive experiments were conducted, showing that the proposed method achieves superior retrieval performance, and a single bit demonstrates interpretability.

**Claims And Evidence:**

From the perspective of cosine similarity, this paper provides an analysis of why retrieval performance is poor in low-bit hashing scenarios within a pairwise setting. The authors also visualize the loss landscape, providing empirical evidence for their theory. Furthermore, the authors claim that the proposed method can generate attribute-aware hash codes. They showcase the results through qualitative comparisons of one bit retrieval results and heat maps. Additionally, the authors explain the attribute diversity of query learning mechanism.

**Essential References Not Discussed:**

No essential references not discussed

**Experimental Designs Or Analyses:**

The authors compare the results of different methods on five fine-grained datasets and present a lot of visualizations. However, a major contribution of the paper is the introduction of auxiliary branches. The key experiments regarding the hyperparameter N are only conducted on one dataset, which makes this part of the experimentation insufficient.

**Methods And Evaluation Criteria:**

The method proposed in the paper is meaningful for the application of fine-grained image retrieval. By focusing on attribute-level information and optimizing low-bit hash codes, it offers a robust and interpretable framework for this task.

**Other Comments Or Suggestions:**

1. The Related Work section on 'Set Prediction and Parallel Decoding' could be divided into two parts.

**Other Strengths And Weaknesses:**

Strengths:
1.The proposed method achieves superior retrieval performance and can generate attribute-aware hash codes.
2.The introduction of the auxiliary branch and the corresponding analysis is interesting.
3.The authors provide many visualization results.

Weaknesses:
1.The paper lacks a comparison with the recent method ConceptHash[1]
2.The paper discusses the limitation of large class numbers with low feature dimensions. However, the experiments related to the hyperparameter N, which are directly relevant to this analysis, were only conducted on one dataset, which is very insufficient
3.The authors provide a detailed description of the motivation behind the auxiliary branch design, but only a brief description is given regarding the specific operations.

Ref:
[1]Ng, K. W., Zhu, X., Song, Y.-Z., and Xiang, T. Concepthash: Interpretable fine-grained hashing via concept discovery. In Proc. IEEE Conf. Comput. Vis. Pattern Recognit. Workshop, pp. 1211–1223, 2024.

**Questions For Authors:**

1. For key questions, please refer to the Weaknesses section.
2. What is the difference between Query Optimization mentioned in the paper and c-vectors optimization in CMBH[1].

Ref:
[1]Chen, Z.-D., Zhao, L.-J., Zhang, Z.-C., Luo, X., and Xu, X.-S. Characteristics matching based hash codes generation for efficient fine-grained image retrieval. In Proc. IEEE Conf. Comput. Vis. Pattern Recognit., pp. 17273–17281, 2024.

**Relation To Broader Scientific Literature:**

The paper presents an advanced method that generates attribute-aware hash codes optimized through queries for large-scale fine-grained image retrieval. Compared to previous works such as AGMH and SEMICON, this method achieves superior retrieval performance

**Theoretical Claims:**

The theoretical proof in the paper seems to be reasonable.

---

> ### Author Rebuttal · Authors · 2025-03-29
>
> **Comment1:** *The paper lacks a comparison with the recent method ConceptHash*
>
> **Reply1:**
> | Method       | bits | CUB-200 | Aircraft | Stanford Cars |
> |--------------|------|---------|----------|---------------|
> | ConceptHash  | 16   | 83.45   | 82.76    | 91.70          |
> |              | 32   | 85.27   | 83.54    |**92.60**|
> |              | 64   | 85.50   | 84.05    | 93.01         |
> | Ours$^+$     | 16   | **84.12**   | **84.16**|**92.12** |
> |              | 32   | **86.02**  | **84.26**| 92.40  |
> |              | 64   |**86.76** | **85.76**|**93.46** |
>
> **Table 2:Comparison of Retrieval Accuracy (%mAP) with ConceptHash.$^+$ denotes our method is trained with classification and uses the ViT-Base backbone.**
>
> Due to the different experimental settings between ConceptHash and our experiment (both in the main paper and appendix). We made adjustments to our own method and re-conducted the experiment. The experimental results are shown in Table 1, demonstrating that our method is comparable to these methods.
>
> ---
>
> **Comment2:** *The paper discusses the limitation of large class numbers with low feature dimensions. However, the experiments related to the hyperparameter N, which are directly relevant to this analysis, were only conducted on one dataset, which is very insufficient.*
>
> **Reply2:**
> |     $N$  | 1      | 2      | 4      | 6      | 8      | 12     |
> |----------|--------|--------|--------|--------|--------|--------|
> | CUB200   | 33.33  | 52.44  | 69.46  | 71.57  | 72.19  | 72.36  |
> | Aircraft | 39.43  | 65.70   | 74.51  | 75.20  | 78.47  | 78.38  |
> | Food101  | 43.79  | 66.16  | 71.91  | 69.67  | 70.69  | 71.08  |
> | NABirds  | 7.48   | 11.01  | 18.88  | 24.17  | 28.13  | 32.08  |
> | VegFru   | 18.16  | 27.76  | 39.08  | 49.66  | 69.76  | 71.68  |
>
> **Table 2: Comparison results of hyperparameter N. Results are based on five commonly used benchmark datasets under the 12-bits setting.**
>
> Thank you for your suggestion. We have provided more experimental results in Table 2. The results on different datasets show that as $N$ increases from 1 to 6, the retrieval performance improves rapidly. Once $N$ exceeds 6, the performance stabilizes, which follows the same trend as the change of mu shown in Figure 3.
>
> ---
>
> **Comment3:** *The authors provide a detailed description of the motivation behind the auxiliary branch design, but only a brief description is given regarding the specific operations.*
>
> **Reply3:**
> For the specific implementation of the auxiliary branch, we provide a detailed description. Given a query $q_i \in \mathbb{R}^d$, we divide $q_i$ evenly into multiple parts. We then perform a circular shift with step sizes of $1, 2, 3, ..., N-1$. For each of the $N-1$ different step sizes, we extend the original $q_i$ into $N-1$ new queries, $\hat{q}_i^j$, where $j = 1, 2, 3, ..., N-1$ indexes the different step sizes and all queries share the same parameters. We refer to this extension process as query transformation. We initialize a total of $k$ learnable queries. After the query transformation operation, each query is passed to the decoder for computation according to Equation 3. Finally, we obtain $\hat{h} \in \mathbb{R}^{N \times k}$, which is optimized using the loss function defined in Equation 6.
>
> **Comment4:** *The Related Work section on 'Set Prediction and Parallel Decoding' could be divided into two parts.*
>
> **Reply4:**
> We sincerely appreciate your valuable suggestion regarding the organization of the related work section. We have restructured this part to improve clarity.
>
> ---
>
> **Questions1:**
> *What is the difference between Query Optimization mentioned in the paper and c-vectors optimization in CMBH？*
>
> **Reply1:**
> Our approach differs from CMBH in the following ways:
> 1. CMBH’s contribution focused on better extracting subtle image features but does not improve the process of generating hash codes. This distinction results in the hash codes generated by CMBH not exhibiting interpretability, while our approach generates hash codes that can indicate whether an image possesses a certain visual attribute.
>
> 2. We primarily follow the pairwise setting, which means that we can not use the classification task for training. In contrast, CMBH’s optimization process is designed based on a classification task, which cannot be used in the pairwise setting.

---

### Official Review · Reviewer_QKWg · 2025-03-11

**Overall Recommendation:** 4

**Summary:**

This work treats the hash learning process as a set prediction problem, using a cross-attention-based decoder to decouple attribute-specific features and further compress them into hash codes. From the perspective of cosine similarity, it argues that large class numbers with low feature dimensions lead to poor retrieval performance in low-bit hash code scenarios, and a query transformation operation is employed as a solution, resulting in a well-optimized set of queries for hash codes generation. The authors provide numerous experimental results that validate the effectiveness of the proposed method.

**Claims And Evidence:**

The paper claims that the proposed method can capture attribute-level information from images. The authors provide single bit retrieval results, demonstrating that the generated hash code can indicate nuanced visual attributes that can distinguish different fine-grained categories. The visualization results of the heatmaps also show that different queries can focus on different parts of the image.

**Essential References Not Discussed:**

The related references are discussed.

**Experimental Designs Or Analyses:**

During the query learning process, no attribute-level annotations were provided. Interestingly, the authors present corresponding quantitative results and visualizations that showcase the attribute-aware characteristics of the generated hash codes and further analyze the diversity of query learning.

**Methods And Evaluation Criteria:**

The proposed method is well-suited for the problem of fine-grained image retrieval. It not only achieves superior retrieval performance but also brings interpretability to the generated hash codes.

**Other Comments Or Suggestions:**

1.Line 58 should use 'demonstrates,' and line 75 should use 'are'.

**Other Strengths And Weaknesses:**

Strengths:

1.The paper utilizes a query learning mechanism with learnable queries to generate attribute-aware hash codes, achieving not only performance improvements but also enhanced interpretability.

2.The explanation from the perspective of cosine similarity is clearly exemplified by Figures 3 and 4. This fresh perspective substantiates its novelty.

3.The experiments in this paper are sufficient, demonstrating significant performance improvements, especially in low-bit scenarios.

Weaknesses:

1.The subtle feature extractor employs a multi-scale framework in conjunction with a multi-head self-attention (MHSA) mechanism. This method is widely adopted in deep learning, particularly for image feature extraction. For instance, architectures like Mobile-Former and ConVit exemplify this trend. While the design is reasonable, it does not offer significant innovation.

**Questions For Authors:**

1.How many layers of cross attention does the decoder part consist of?

**Relation To Broader Scientific Literature:**

The innovation of this paper is reflected in its practicality, as it combines semantic decoupling of fine-grained hashing with a lightweight model. This approach effectively addresses the performance limitations of existing methods (such as ExchNet and SEMICON) in low-bit scenarios. By introducing a learnable query mechanism, the proposed method differentiates from the attribute-aware learning framework of A²-NET, achieving automated attribute decoupling of complex global image features.

**Theoretical Claims:**

The proof of the lower bound for μ in the paper seems to be free of issues.

---

> ### Author Rebuttal · Authors · 2025-03-29
>
> **Comment1:** *The subtle feature extractor employs a multi-scale framework in conjunction with a multi-head self-attention (MHSA) mechanism. This method is widely adopted in deep learning, particularly for image feature extraction. For instance, architectures like Mobile-Former and ConVit exemplify this trend. While the design is reasonable, it does not offer significant innovation.*
>
> **Reply1:**
> We would like to emphasize that:
>
> 1. We model the hash problem as a set prediction problem, where each element in the set represents a bit of the hash code that can indicate a visual attribute. Specifically, our method uses $ k $ learnable queries to directly decouple distinguishable visual attributes from complex feature representations to generate the $ k $-bit hash code.
>
> 2. Previous fine-grained retrieval methods have overlooked the low-bit problem. We provide an analysis from the perspective of cosine similarity and design a query transformation strategy that effectively alleviates this issue.
>
> 3. Regarding the design of the subtle feature extractor, our main objective is to use a lightweight and simple strategy to extract fine-grained features. Overly complex feature extraction modules also lead to larger model parameters and greater computational overhead. Some previous methods have primarily focused on the design of this module, while neglecting improvements in the hash code generation process.
>
> Overall, the first two key contributions have never been explored in previous work. At the same time, using our subtle feature extractor makes the method more lightweight and reduces computational overhead..
>
> ---
>
> **Comment2:** *Line 58 should use 'demonstrates,' and line 75 should use 'are'.*
>
> **Reply2:**
> We sincerely appreciate your thorough review. We have carefully addressed the identified issues and conducted further proofreading of the manuscript to ensure its quality.
>
> ---
>
> **Questions1:**
> *How many layers of cross attention does the decoder part consist of?*
>
> **Reply1:**
> The decoder consists of a single layer. Its lightweight design ensures that the introduction of auxiliary branches does not incur significant computational overhead.

---

> > ### Comment · Reviewer_QKWg · 2025-04-02
> >
> > I have carefully read the authors' rebuttal and the feedback has well addressed my questions and concerns. Thus, I would like to raise the score to 4. Thanks.

---

### Official Review · Reviewer_ATUG · 2025-03-13

**Overall Recommendation:** 3

**Summary:**

This paper presents a query optimization-based attribute-aware hash code generation method. First, a hybrid convolution and attention structure is utilized to obtain rich representations. Second, unlike other works that simply use fully connected layers to generate hash codes, this paper leverages a decoder and a set of learnable queries to automatically decouple different attribute features. Third, the paper incorporates an auxiliary branch to help alleviate the challenges of complex landscape optimization. Quantitative experimental results demonstrate the high retrieval performance of the method, while qualitative results show that the learned bits exhibit visual attribute-level interpretability. Additionally, the method is relatively lightweight, resulting in low computational overhead in practical applications.

**Claims And Evidence:**

The claims made in the submission are clearly supported by convincing evidence. The author provides a well-rounded and methodologically sound approach. The motivation and the method of this paper are clearly described, and the extensive experimental results offer rigorous validation, demonstrating the superior efficacy and robustness of the proposed method across diverse datasets and scenarios.

**Essential References Not Discussed:**

NA

**Experimental Designs Or Analyses:**

The authors conducted extensive experiments to compare the performance of the proposed method with other methods. Additionally, the authors performed experimental analysis on the method itself, which helps readers better understand the proposed method.

**Methods And Evaluation Criteria:**

The large-scale fine-grained retrieval problem studied in this paper is a fundamental and practical task, and the proposed method can be treated as an effective solution.

**Other Comments Or Suggestions:**

1. Figure 7 could provide a more detailed description.

**Other Strengths And Weaknesses:**

Strengths:

1. This paper is well written and easy to follow.

2. The motivation is clear, and the method is reasonable. The generated hash codes are associated with distinguishable attribute-level information, which brings interpretability to the retrieval process. Qualitative experimental results show that the proposed method offers interpretability.

3. The experimental design is well-structured, and Section 4.3 provides a thorough analysis of the experiments, offering a comprehensive evaluation of the proposed method.

4. The proposed method demonstrates strong performance while requiring less computational overhead.

Weaknesses:

1. One of the main contributions of the paper is the introduction of an additional auxiliary branch. However, the description of this operation is too brief and does not sufficiently clarify its specific implementation. It is recommended that the authors provide a more detailed explanation of the auxiliary branch, including the interaction between queries and features, so that readers can better understand the significance and practical implications of this contribution.

**Questions For Authors:**

What is the relationship between the two decoders in Figure 4?

**Relation To Broader Scientific Literature:**

This paper proposes an efficient attribute-aware hash code generation method. Unlike the stepwise attention proposed by AGMH for better feature extraction, this paper uses a decoder to decouple the extracted features, allowing the generated hash bits to keep key information from the images. This idea contributes a solution to the field of fine-grained image retrieval. Additionally, the introduction of an auxiliary branch is quite interesting. It incurs only a small amount of extra computational overhead, helping the model to alleviate the challenges of optimization in low-bit scenarios while enhancing performance.

**Theoretical Claims:**

I have checked the proofs concerning the theoretical claims. The analysis and proofs are both reasonable.

---

> ### Author Rebuttal · Authors · 2025-03-28
>
> **Comment1:** *One of the main contributions of the paper is the introduction of an additional auxiliary branch. However, the description of this operation is too brief and does not sufficiently clarify its specific implementation. It is recommended that the authors provide a more detailed explanation of the auxiliary branch, including the interaction between queries and features, so that readers can better understand the significance and practical implications of this contribution.*
>
> **Reply1:**
> We sincerely appreciate your valuable feedback. Without the auxiliary branches, the learnable queries interact with the extracted features in a fixed manner. For instance, each query can be split into $N$ segments. When $N=2$, the first segment consistently interacts with features corresponding to the first half of the image channels, while the second segment interacts with features from the latter half. However, when an auxiliary branch is introduced, the interaction pattern undergoes a significant change. In the case where $N=2$, the first segment of the query can interact with features from either the first half or the latter half of the image channels, and the same applies to the second segment. The introduction of auxiliary branches expands the receptive range of queries across the channel dimension. Since different channels typically encode distinct semantic information or visual features, this enhancement allows the optimized queries to better capture visual attributes across different images.
>
> ---
>
> **Comment2:** *Figure 7 could provide a more detailed description.*
>
> **Reply2:**
> We sincerely appreciate your valuable feedback. In Figure 7, we present the visualization results of heatmaps across different datasets. Each row corresponds to a distinct learnable attribute query ($q_i \in \mathbb{R}^d$). The heatmaps generated by different $q_i$ demonstrate that the learned queries effectively focus on different parts of the objects. For instance, on $CUB200$ dataset, certain queries attend to the body, while others focus on the beak of a bird. On $Aircraft$ dataset, some queries focus on the wings, while others focus on the tail of the airplane. These visualizations qualitatively illustrate that our proposed query learning achieves strong interpretability.
>
>
> ---
>
> **Questions1:**
> *What is the relationship between the two decoders in Figure 4?*
>
> **Reply1:**
> Figure 4 shows the visualized results of the loss landscape under different conditions.
> In Figure 5, the two decoders share the same weights.

---

### Decision · Program_Chairs · 2025-05-01

**Decision:**

Accept (poster)

**Comment:**

This paper proposes a query optimization-based method for generating attribute-aware hash codes in fine-grained image retrieval. It combines a hybrid CNN-attention encoder with a learnable query decoder and an auxiliary branch to enhance optimization, especially under low-bit settings. The method is well-motivated and tailored for fine-grained tasks, demonstrating good accuracy and good interpretability across multiple benchmarks.

The four reviewers (2 Accept, 2 Weak Accept) agree the work is solid, with clear contributions, effective experiments, and meaningful insights. Minor improvements could include clarifying the auxiliary branch and adding more recent baselines. Overall, a promising and well-executed paper.